# Hedging Long-Dated Oil Futures and Options Using Short-Dated Securities—Revisiting Metallgesellschaft

James S. Doran [1] and Ehud I. Ronn [2],*

1    School of Banking and Finance, UNSW Business School, Sydney, NSW 2052, Australia; j.doran@unsw.edu.au
2    Department of Finance, McCombs School of Business, University of Texas at Austin,
     2100 Speedway Stop B6600, Austin, TX 78712-1276, USA
*    Correspondence: eronn@mail.utexas.edu; Tel.: +1-(512)-471-5853; Fax: +1-(512)-471-5073

**Abstract:** Since the collapse of the Metallgesellschaft AG due to hedging losses in 1993, energy practitioners have been concerned with the ability to hedge long-dated linear and non-linear oil liabilities with short-dated futures and options. This paper identifies a model-free non-parametric approach to extrapolating futures prices and implied volatilities. When we expand the analysis to implementing hedge portfolios for long-dated futures or option contracts over the time period 2007–2017, we utilize the useful benchmark of hedge ratios arising from Schwartz and Smith. With respect to the empirical consequences of hedging long-dated futures and options with their short-dated counterparts, we find that the long-term tracking errors are, on average, quite close to zero, but there is increasing risk entailed in attempting to do so, as the hedge-tracking errors for both futures and option contracts increase with time-to-maturity.

**Keywords:** hedging; long-dated; oil futures; option contracts



**JEL Classification:** G12—asset pricing; G13—contingent pricing

## 1. Introduction

Edwards and Canter (1995) summarize the misfortunes of the German conglomerate Metallgesellschaft AG:

> "In late 1993 and early 1994 MG Corporation, the U. S. subsidiary of Germany's 14th largest industrial firm Metallgesellschaft AG reported staggering losses on its positions in energy futures and swaps. Only a massive $1.9 billion rescue operation by 150 German and international banks kept MG from going into bankruptcy".

The website, `wikipedia.org`, accessed on 12 June 2021, added that "Subsequently, the spot price increased and the company suffered even greater losses covering its customer commitments".

Although we do not pursue that matter here, one of the important economic issues raised with respect to Metallgesellschaft was the issue of the substantial credit risk that faced the company: hedging non-marked to market OTC commitments with futures contracts. It has been argued that, irrespective of whether their hedging policies would have been effective, the company would have confronted the risk of defaults among its relatively poorly funded private counterparties who could not have afforded to pay higher wholesale prices if the market prices for their products had fallen significantly.

In the aftermath of these substantial hedging losses, discussions took place in the academic literature regarding the source of the hedging issues that Metallgesellschaft encountered. Wahrenburg (1996) summarized it by noting "On the one hand, Culp and Miller (1994, 1995a, 1995b) argue that the strategy of Metallgesellschaft was basically sound and effectively reduced MGRMs oil price risk. On the other hand, a number of

authors like Edwards and Canter (1995) and Mello and Parsons (1995) argue that instead of reducing its oil price risk, MGRM actually increased risk by using a grossly oversized hedge position". In another earlier paper, Ronn and Xuan (1998) were able to confirm a "one-factor hedge using the near-month performs worse than a no-hedge policy, consistent with the conjecture of Mello and Parsons (1995)". More recently, Manley and Shetty (2019) reviewed the Metallgesellschaft use of a "rolling stack of short-dated futures, [concluding] when cash flows matter, the rolling stack may be worse than no hedge at all".

Research has since addressed the matter from theoretical perspectives, including the works of Brennan and Crew (1997); Pirrong (1997); Lien and Tse (2002); Veld-Merkoulova and De Roon (2003); Bühler et al. (2004); and Saha (2011). In particular, Neuberger (1999) "analyzes the problem facing an agent who has a long-term commodity supply commitment and who wishes to hedge that commitment using short-maturity commodity futures contracts... The optimal hedging strategy is characterized in a world where contracts of several different maturities coexist. The strategy is independent both of the agent's risk aversion and, under certain conditions, of beliefs about expected returns from holding futures contracts".

Empirical research has dealt with this issue using distinct approaches. Thus, Ripple and Moosa (2007) contrast a single-factor hedging using "the near-month contract and those resulting from the use of a more distant (6-month) contract. The results show that futures hedging is more effective when the near-month contract is used". Cheng et al. (2016) focus exclusively on the hedging of long-dated option contracts. Guo (2017) considers risk measures for oil, natural gas, gasoil, heating and electricity futures contracts. Shiraya and Takahashi (2012) and Lautier and Galli (2010) exclusively consider long-term contracts using prices from the oil and (for the former paper) copper markets. The work of Alizadeh et al. (2004) addresses the issue of hedging international oil prices. Glasserman (2001) focuses on the matter of "shortfall" in hedging the risk of long-dated futures contracts. Finally, Carbonez et al. (2011) address the issue of hedging long-dated contracts in the agricultural sector.

In contrast, in the current paper, we take a model-free, non-parametric approach to implementing the extrapolation of long-dated commitments with shorter-maturity securities, pointing to its potentials as well as pitfalls. In contrast to our predecessors, we considered analysis of both futures and option contracts, utilizing the convenient hedge ratios implicit in the Schwartz and Smith (2000) paper.

A component of the motivation for our paper pertains to the matter of numerous agents' need to hedge long-dated futures contracts—i.e., beyond those of the "liquidity term" provided by active futures contracts. If such liquidity terms are defined as $T \leq 2$ years, it is clear that real assets in the oil sector have economic lives well beyond these liquidity terms. Thus, the issue of hedging such long-dated assets is of considerable relevance to these agents. By extension, this also applies to commercial agents, such as financial intermediaries, who are tasked with the issue of hedging long-dated exposures. Our interest is, therefore, in both the extrapolation of futures prices beyond the liquidity term, as well as the issue of hedging such exposures.

Using data covering the period 2007–2017, our work in this research can be divided into two parts. The first is the model-free extrapolation of long-dated futures prices. The second part deals with the issue of hedging long-dated exposures: In this section, we use the above-referenced two-factor model provided by Schwartz and Smith (2000). As described in detail in Section 4.1, the model provides the requisite number, two, of liquid futures contracts to hedge long-dated contracts. As specified in Section 4.2, this model also indicates how to expand the hedge portfolio when seeking to hedge long-dated, volatility-sensitive option contracts.

With respect to the empirical conclusions of our work pertaining to hedging long-dated futures and options with their short-dated counterparts, we find that the long-term tracking errors are, on average, quite close to zero, but there is increasing risk entailed in

attempting to do so, as the hedge-tracking errors for both futures and option contracts increase with time-to-maturity.

The importance of the oil futures market is clear: It is the most liquid futures contract in the commodity arena.[1] Since liquidity in long-term contracts fluctuates due to various economic forces, the ability to hedge long-dated contracts remains ever-critical. This paper is now organized as follows. Section 2 presents the relevant crude-oil futures and options data we utilize. Section 3 focuses on the purely numerical issues pertaining to the extrapolation of futures and option contracts to more-distant maturities. In turn, Section 4 proceeds with the details of the empirical hedging tests to which we subject long-dated futures and options data. Section 4 summarizes the results of this research.

## 2. Data

Based on the considerable oil futures and options data included in the Bloomberg platform, we use the prices of WTI crude-oil futures and options data on the NYMEX. The time period $t$ covered by the data is 29 November 2007 through 30 June 2017. While futures prices are observed as far out as nine years, we choose the first 24 months as the range of traded contracts that can be easily traded (the maturities are denoted by the notation $T \leq 2$). With respect to the options, Bloomberg reports the at-the-money implied volatilities, for which the notation is $\sigma_{tT}$. While these volatilities cover the same maturity range as their corresponding futures contracts, they are not as fully populated—that is, there are some expiration dates $T \leq 2$ for which the implied volatilities were not reported on some of the dates $t$.

In the data, it is critical to distinguish between prices of futures and options with less than two years to maturity, and those beyond. While we do have published prices for maturities up to eight years, these longer-dated instruments are quite illiquid. Thus, we posit the need to hedge longer-dated exposures using short-dated instruments $T \leq 2$ years.

## 3. Extrapolating the Prices of Oil Futures and Options Contracts

Assume the following notation:

$F_{tT}$ = Futures contract for maturity $T$ on any calendar date $t$. The subscript $t$ may be suppressed without ambiguity or loss of generality.

$\sigma_{tT}$ = At-the-money implied volatility for expiration date $T$ on calendar date $t$.

Since we have observable data available for futures and (sometimes partially) for options out to $T \leq 2$, the assumption is that these maturities and expiration dates are the "short-dated" ones. When we seek to test our ability to hedge exposures further out, we examine our empirical results for the "longer-dates" exposures out six *more* years, that is, for maturities out to $T = 3, 4, \ldots, 8$.

### 3.1. Extrapolating Futures Prices

We begin by considering a model and tests for extrapolating and hedging long-dated futures contracts using their shorter-maturity counterparts. Thus, at each date $t$ using observable futures contract prices out to $T \leq 2$, consider fitting futures prices using a cubic function:

$$F_{tT} = a_t + b_{1t}T + b_{2t}T^2 + b_{3t}T^3 \qquad (1)$$

where the parameters to be estimated at each date $t$ are $\boldsymbol{\alpha}_u \equiv \{a_t, b_{1t}, b_{2t}, b_{3t}\}$.

We designated the above an "Unconstrained" $\boldsymbol{\alpha}_u$ estimation, as the estimated parameters are unconstrained, and the resulting estimation can be achieved via standard linear regression. In contrast, we have a second "Constrained" estimation procedure, whereby on each estimation date $t$, we impose two constraints on the parameter set—the first,

Equation (2), to induce smoothness at $T = 8$, and the second, Equation (3), to ensure the fitted $F_{T=8}$ is non-negative: suppressing the subscript $t$,

$$\left. \frac{\partial F_T}{\partial T} \right|_{T=8} = b_1 + 2b_2T + 3b_3T^2$$

$$= b_1 + 16\,b_2 + 192\,b_3 = 0 \tag{2}$$

$$F_{T=8} = a + 8\,b_1 + 64\,b_2 + 512\,b_3 \geq 0 \tag{3}$$

we proceed to estimate these parameters by solving the minimization problem:[2]

$$\min_{\{a,\,b_1,\,b_2,\,b_3\}} \left[ \sum_T \left( F_T - a - b_1T - b_2T^2 - b_3T^3 \right)^2 + 1000(b_1 + 16\,b_2 + 192\,b_3)^2 \right]$$

Especially for the "constrained results" in Table 1's second panel, subsequent to Table 3 we comment on the magnitude of the Min/Max values. The top panel's result from exponentiation, since it is clearly infeasible for tracking errors to exceed futures prices.

**Table 1.** Futures' estimation errors by time to maturity $T$.

| Unconstrained | | | | | | | |
|---|---|---|---|---|---|---|---|
| | $T = 2$ | $T = 3$ | $T = 4$ | $T = 5$ | $T = 6$ | $T = 7$ | $T = 8$ |
| Average | −$0.01 | $0.01 | $0.02 | $0.03 | $0.04 | $0.04 | $0.04 |
| Std. Dev. | $0.10 | $1.89 | $7.97 | $20.91 | $43.16 | $77.13 | $125.24 |
| Max | $1.05 | $25.65 | $105.31 | $290.93 | $611.94 | $1103.28 | $1799.90 |
| Min | −$0.81 | −$22.78 | −$112.75 | −$310.70 | −$652.80 | −$1176.17 | −$1917.97 |
| Constrained | | | | | | | |
| | $T = 2$ | $T = 3$ | $T = 4$ | $T = 5$ | $T = 6$ | $T = 7$ | $T = 8$ |
| Average | −$0.01 | $0.00 | $0.00 | $0.01 | $0.01 | $0.01 | $0.01 |
| Std. Dev. | $0.13 | $0.92 | $1.83 | $2.77 | $3.64 | $4.28 | $4.53 |
| Max | $1.60 | $12.87 | $23.65 | $34.05 | $43.21 | $49.82 | $52.41 |
| Min | −$1.65 | −$12.34 | −$22.63 | −$32.62 | −$41.45 | −$47.85 | −$50.36 |

*3.2. Extrapolating Implied Volatility*

Analogous to the procedures we implement for the futures contracts, we now extend these results to the calibration of the implied volatilities:

$$\sigma_{tT}^2 = A_t + B_{1t}T + B_{2t}T^2 + B_{3t}T^3 \tag{4}$$

$$\left. \frac{\partial \sigma_T^2}{\partial T} \right|_{T=8} = B_1 + 2B_2T + 3B_3T = b_1 + 16\,b_2 + 192\,b_3 = 0 \tag{5}$$

where the parameters to be estimated at each date $t$ are $\boldsymbol{\beta}_c \equiv \{A_t,\, B_{1t},\, B_{2t},\, B_{3t}\}$.

The results of this extrapolation methodology are reported in Table 2.

**Table 2.** Implied volatility estimation errors by time to maturity $T$.

| Unconstrained | | | | | | | |
|---|---|---|---|---|---|---|---|
| | $T=2$ | $T=3$ | $T=4$ | $T=5$ | $T=6$ | $T=7$ | $T=8$ |
| Average | 0.00 | 0.04 | 0.08 | 0.11 | 0.14 | 0.16 | 0.18 |
| Std. Dev. | 0.15 | 1.92 | 3.69 | 5.77 | 8.59 | 12.63 | 18.30 |
| Max | 1.04 | 16.06 | 29.43 | 49.42 | 95.62 | 164.57 | 260.98 |
| Min | $-1.04$ | $-17.38$ | $-32.02$ | $-45.30$ | $-88.24$ | $-156.54$ | $-252.81$ |
| Constrained | | | | | | | |
| | $T=2$ | $T=3$ | $T=4$ | $T=5$ | $T=6$ | $T=7$ | $T=8$ |
| Average | 0.00 | 0.01 | 0.01 | 0.01 | 0.01 | 0.01 | 0.01 |
| Std. Dev. | 0.02 | 0.08 | 0.13 | 0.16 | 0.19 | 0.20 | 0.21 |
| Max | 0.19 | 0.58 | 0.87 | 1.11 | 1.28 | 1.39 | 1.44 |
| Min | $-0.16$ | $-0.44$ | $-0.69$ | $-0.89$ | $-1.03$ | $-1.13$ | $-1.17$ |

*3.3. Summary*

The straightforward conclusion of this section is the ease and feasibility of obtaining a proxy for long-dated futures and option contracts using their short-term counterparts, especially when one takes advantage of the smoothing of these curves using the asymptotic condition imposed on the slopes of the futures and option contracts.[3] The relevance of this is not merely for financial assets. Here, it is important to reiterate that the valuation of real assets in the oil and natgas industries relies critically on the prices of futures contracts—and wherever there are embedded optionalities, those of option contracts.

**4. Hedging Long-Dated Futures and Options with Short-Dated Maturities**

*4.1. Hedging Long-Dated Futures with Short-Dated Futures*

Taking advantage of the $c_T \cong \dfrac{\sigma_T \sqrt{T} \exp\{-rT\}}{\sqrt{2\pi}} F_T$ Brenner and Subrahmanyam (1988) approximation for an ATM option, under the two-factor Schwartz–Smith model, each futures contract price $F_T$ satisfies

$$
\begin{aligned}
d \ln F_T &= \exp\{-\kappa T\} d\chi_t + d\xi_t \\
\ln F_T &= \xi_0 + e^{-\kappa T}\chi_0 + \left(1 - e^{-\kappa T}\right)\chi_1 + \mu_\xi T + \frac{1}{2}\sigma_T^2 T \\
\sigma_T^2 T &= \left(1 - e^{-2\kappa T}\right)\frac{\sigma_\chi^2}{2\kappa} + \sigma_\xi^2 T + 2\left(1 - e^{-\kappa T}\right)\frac{\rho_{\chi\xi}\sigma_\chi\sigma_\xi}{\kappa} \\
c_T &= \frac{\sigma_T \sqrt{T} \exp\{-rT\}}{\sqrt{2\pi}} F_T
\end{aligned}
\tag{6}
$$

where $c_T$ are the prices of ATM-Forward (call or put) options.[4]

We perceive at least two advantages to this application of the Schwartz and Smith (2000) model:

1. The model is a two-factor model, one of which is mean-reverting. This is not merely a theoretical virtue, but it has an important implication that conforms nicely to the empirical regularities. Specifically, consider the model's implications for the term structure of volatility (TSOV), that is, the graphical depiction of implied vols as a function of time.

    The mean-reverting factor gives rise to a downward-sloping TSOV, as the effect of the mean-reverting factor "dies out'.' The second, non-mean reverting factor gives rise to a positive lower bound to long-dated volatilities. Both of these attributes, a declining TSOV and a positive asymptotic behavior, are reflected in the stylized facts that we observe for both historical as well as implied volatilities. These are properly identified in the enclosed Figure 1, excerpted from Schwartz (1997).

2. The second advantage is that it identifies the number of factors we should use to hedge exposures.

Specifically, the two-factor Schwartz–Smith specification (6) permits us to solve for the correct hedge portfolio for an arbitrary futures contract of maturity $T$. Rewriting (6) for maturities $T = 1$ and $T = \tau$ produces two equations in the two unknowns $d\chi_t$ and $d\xi_t$. Solving $d\chi_t$ and $d\xi_t$ and substituting into (6) provides the expression for $d\ln F_T$ as a function of $d\ln F_\tau$ and $d\ln F_1$ :

$$
\begin{aligned}
d\ln F_T &= [(\exp\{-\kappa T\} - \exp\{-\kappa\})/(\exp\{-\tau\kappa\} - \exp\{-\kappa\})]d\ln F_\tau \\
&\quad + [(\exp\{-\tau\kappa\} - \exp\{-\kappa T\})/(\exp\{-\tau\kappa\} - \exp\{-\kappa\})]d\ln F_1 \\
&\equiv w_\tau\, d\ln F_\tau + w_1\, d\ln F_1
\end{aligned}
\tag{7}
$$

In turn, this hedge portfolio (7) merits three comments:

1. The weights $w_\tau$ and $w_1$ sum to unity, but they both will not be positive if $T > \tau > 1$.
2. For purposes of our empirical hedging tests, we choose $\tau = 2$. For the $T > 2$ cases whose hedging we will examine, the hedge portfolio takes a long position in the $T = 2$ futures contract and a short position in the $T = 1$ contract. The long position in the $T = 2$ futures is intuitively appealing, since for $T > 2$, the $T = 2$ contract is the one "closest to" the $T > 2$ contract. Moreover, the short position in the $T = 1$ contract may be of hedging assistance when the futures curve changes slope.
3. Since there are contracts not identically equal to $T = 2$ and $T = 1$ years to maturity, we use the fitted prices of these two contracts on subsequent days to compute the hedging errors. Thus, using $w_\tau\, d\ln F_\tau$ and $w_1\, d\ln F_1$ from (7), the hedging errors are computed as $(w_\tau\, d\ln F_\tau)F_\tau + (w_1\, d\ln F_1)F_1$ less the change in the fitted value for maturity $T$, which we compute for $T$ values of 3, 4, . . . , 8.

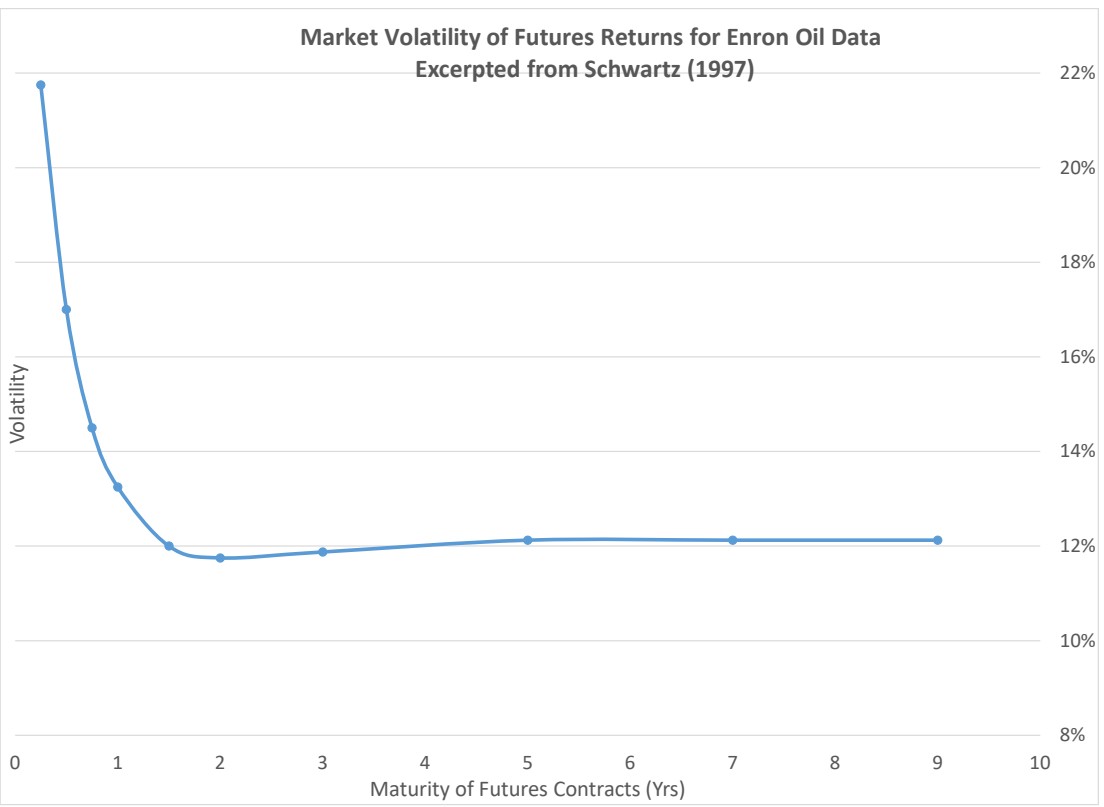

**Figure 1.** Market Volatility of Futures Returns for Enron Oil Data Enron Oil Data Excerpted from Schwartz (1997).

Contrasting $d \ln F_T$ with the actual log change will give rise to an estimate of the tracking errors. For the results of Table 3 only, we use an *ex-post* value of $\kappa = 0.604$, which minimizes the average tracking error:

From Table 3:

1.  On average, the tracking error is very small, signifying a relative success in hedging adjoining-dates price changes.
2.  Unsurprisingly, the std. dev. of the tracking-hedge error increases with maturity.
3.  The Max and Min numbers are misleading, in the sense they reflect the worst-case scenarios encountered during the height of the financial crisis. Combining the average with the std. dev. numbers, most of the time the numbers are well confined to, at most, several dollars for the longer maturities.

**Table 3.** Tracking error for futures contracts.

|  | $T = 3$ | $T = 4$ | $T = 5$ | $T = 6$ | $T = 7$ | $T = 8$ |
|---|---|---|---|---|---|---|
| Average | \$(0.001) | \$0.003 | \$0.006 | \$0.009 | \$0.010 | \$0.011 |
| Std. Dev. | \$0.925 | \$1.826 | \$2.773 | \$3.639 | \$4.276 | \$4.525 |
| Min | \$(12.34) | \$(22.63) | \$(32.62) | \$(41.45) | \$(47.85) | \$(50.36) |
| Max | \$12.87 | \$23.65 | \$34.05 | \$43.21 | \$49.82 | \$52.41 |

### 4.2. Hedging Long-Dated Options with Short-Dated Futures and Options

For options, we use a hedge portfolio composed of the $T = 2$ futures and option contracts. In other words, we have [5]

$$dc_8 = \frac{\partial c_8}{\partial \sigma_8} \frac{\partial \sigma_8}{\partial \sigma_2} \frac{\partial \sigma_2}{\partial c_2} \, dc_2 + \frac{\partial c_8}{\partial F_8} \frac{\partial F_8}{\partial F_2} \, dF_2, \tag{8}$$

whose analytics are laid out in Table 4:

**Table 4.** Hedge portfolio for $T = 8$ options.

| Derivative | Analytics | Rationale |
|---|---|---|
| $\dfrac{\partial c_8}{\partial \sigma_8} =$ | $\dfrac{\sqrt{8} \exp\{-8r\} F_8}{\sqrt{2\pi}}$ | Brenner and Subrahmanyam (1988) approximation |
| $\dfrac{\partial \sigma_8}{\partial \sigma_2} =$ | $\dfrac{\sqrt{A + B_1 * 8 + B_2 * 64 + B_3 * 512}}{\sqrt{A + B_1 * 2 + B_2 * 4 + B_3 * 8}}$ | Linear approximation between the $T = 2$ and $T = 8$ vols |
| $\dfrac{\partial \sigma_2}{\partial c_2} =$ | $\dfrac{\sqrt{2\pi}}{\left(\sqrt{2} * \exp\{-2r\}\right)} F_2$ | Brenner and Subrahmanyam (1988) approximation |
| $\dfrac{\partial c_8}{\partial F_8} =$ | $\dfrac{\sqrt{A + B_1 * 8 + B_2 * 64 + B_3 * 512} * \sqrt{8} * \exp\{-8r\}}{\sqrt{2\pi}}$ | Brenner and Subrahmanyam (1988) approximation |
| $\dfrac{\partial F_8}{\partial F_2} =$ | $\dfrac{\sqrt{a + b1 * 8 + b2 * 64 + b3 * 512}}{\sqrt{a + b1 * 2 + b2 * 4 + b3 * 8}}$ | Linear approximation between the $T = 2$ and $T = 8$ futures prices cubic fit |

For the results of Table 5, we use a value of $\kappa = 0.825$ :

**Table 5.** Tracking error for option contracts.

|  | $T = 2$ | $T = 3$ | $T = 4$ | $T = 5$ | $T = 6$ | $T = 7$ | $T = 8$ |
|---|---|---|---|---|---|---|---|
| Average | $0.000 | $0.006 | $0.009 | $0.011 | $0.012 | $0.012 | $0.013 |
| Std. Dev. | $0.016 | $0.083 | $0.130 | $0.163 | $0.187 | $0.202 | $0.209 |
| Min | $(0.159) | $(0.440) | $(0.689) | $(0.886) | $(1.035) | $(1.130) | $(1.165) |
| Max | $0.186 | $0.581 | $0.874 | $1.106 | $1.280 | $1.394 | $1.435 |

From Table 5:

1. On average, the tracking error is very small, signifying a relative success in hedging adjoining-dates option-price changes. This has the merit of rebalancing options no more frequently than daily, which can be of considerable importance in light of options' bid-ask spreads.
2. Unsurprisingly, the std. dev. of the tracking-hedge error increases with maturity. The leveling off of std. dev. between years 7 and 8 is possibly reflective of the nature of options' so-called "term structure of volatility". It generally levels off after approx. two years.
3. The Max and Min numbers are considerably lower than they are for futures contracts, but this is, in turn, a consequence of oil options prices being far lower than their corresponding futures prices.

*4.3. Summary*

This section has performed two types of analyses. In the first, we sought to examine how closely we can match the prices of futures and option contracts beyond the liquidity horizon of two-years maturities. In both cases, we focused our attention on the so-called "constrained" case, which imposed smoothness and continuity conditions in requiring the estimated futures prices and implied volatilities to level out after eight years. This has resulted in estimation errors, which we computed for maturities extending from three years to eight.

For both futures and option contracts, the results demonstrated that, while the average errors were de minimus, the standard deviations increased with time to maturity, as did the maximal and minimal errors.

The second set determined the measure of tracking errors in hedging futures contracts with two shorter-dated maturities, as well as hedging options with short-dated options and futures contracts. Again, we found minimal average errors, but standard deviations that intuitively increased with time to maturity.

**5. Conclusions**

Dating back to the early 1990s, if not earlier, the importance of being able to hedge long-dated futures and option contracts with their liquid counterparts was, and is, of importance to any user of lond-dated contracts: those valuing real options, those seeking to hedge long-dated exposures and, of course, those offering long-dated contracts to their commercial counterparts. This paper has demonstrated both the feasibility and challenge in seeking to do this. While, on average, the long-term errors are quite close to zero, there is increasing risk entailed in attempting to do so, as the estimation errors for both futures and option contracts increase with time-to-maturity.

In Section 3's extrapolation section, we used a model-free approach to demonstrate the feasibility of model-free extrapolation accompanied by smoothing conditions imposed on the slopes of both futures and option contracts. This resulted in low-average errors but increasing variability.

Section 4 proceeded beyond extrapolation to the matter of two-factor hedging of both futures and option contrasts as implied by the Schwartz and Smith (2000) model. While the use of a specific model may be deemed constraining, the selection of this specific model as the guidepost to hedging was dually motivated—by the model's two-factor parsimony as

well as its consistency with the "term structure of volatility" (TSOV) empirical regularity observed in most commodity futures contracts. This model's observed TSOV bears the so-called Samuelson effect (whereby the volatilities of long-dated futures contracts are lower than those of short-term futures), while at the same time observing consistency with a volatility asymptote that is strictly positive.[6] in Figure 1 above. The model's results are borne out empirically by its low average error while admittedly accompanied by the result also observed in Section 3's extrapolation, i.e., an increasing standard deviation as the maturity of the hedged instrument increases.

While there are several conceivable directions by which to extend this work, one of the interesting ones reverts back to the matter originally raised in the Introduction: The important economic issue pertaining to the substantial credit risk Metallgesellschaft AG faced: How optimally to hedge non-marked to market OTC commitments with futures contracts.

**Author Contributions:** Conceptualization, E.I.R.: Data curation, J.S.D.: Formal analysis, E.I.R.: Investigation, J.S.D.: Methodology, J.S.D. and E.I.R.: Software, J.S.D.: Validation, J.S.D. and Ehud I. Ronn: Writing—original draft, E.I.R.: Writing—review, editing, E.I.R. All authors have read and agreed to the published version of the manuscript.

**Funding:** This research received no external funding.

**Institutional Review Board Statement:** The Research did not involve human subjects.

**Data Availability Statement:** The data utilized in this study is available on the Bloomberg LLP platform: That platform provides the prices of WTI crude-oil futures and options data on the NYMEX. The time period *t* covered by the data is Nov. 29, 2007 through 30 June 2017.

**Conflicts of Interest:** The authors declare no conflict of interest.

## Notes

1     Per `investopedia.com`, "Crude oil leads the pack as the most liquid commodity futures".

2     Especially for the "constrained results" in Table 1's second panel, subsequent to Table 3 we comment on the magnitude of the Min/Max values. The top panel's result from exponentiation, since it is clearly infeasible for tracking errors to exceed futures prices.

3     Quants will recognize the derivative conditions (2) and (5) as variants of the familiar "smooth-pasting" condition from option theory.

4     Note the expression $\frac{1}{2}\sigma_T^2 T$ in Equation (1) follows from the property of the LogNormal distribution: If $\ln x \sim N(\mu, \sigma^2)$, then the log expectations of $x$ is given by $\ln E(x) = \mu + \frac{1}{2}\sigma^2$.

5     Whereas Equation (8) is stated for $T = 8$, it applies equally and analogously for the other values of $T$, $T = 3, 4, \ldots, 7$.

6     This is demonstrated empirically

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
