# Peer review of "Hedging Long-Dated Oil Futures and Options Using Short-Dated Securities—Revisiting Metallgesellschaft"

_jrfm, doi:10.3390/jrfm14080379_

Round 1
Reviewer 1 Report
The topic is very niche and specific. This could be a good reference for related researchers. The authors should specific more clearly how the data were obtained. Moreover, a more detail discussion should be provided on the background of this study and providing a theoretical framework to help readers.
Reviewer 2 Report
The paper covers an interesting and important topic of the application of short-term derivatives in order to hedge long-term liabilities, in the context of the energy and oil markets. The overall quality of the paper is high - the research methods were selected and applied correctly; the conlusions are grounded in the analysis. I have some minor comments.
- Both abstract and introduction should include some brief overview of the time of the analysis and paper's contributions.
- The references seem outdated - more recent publications should be added.
- The final section should be expanded and include more in-depth discussion of the issues such as limitations of the analysis, comparisons to the results of the previous studies, implications of the analysis, directions for the future research.
Reviewer 3 Report
The article submitted for consideration is devoted to a dareamless non-
parametric approach to extrapolation of futures prices and alleged volatility. The
presented document demonstrates both feasibility and the complexity of attempts
to do this: although on average, long-term errors are quite close to zero, an attempt
to make it entails an increasing risk, because evaluation errors for both futures and
option contracts increase over time until repayment .
In general, the article is interesting, which has useful information. There are
no fundamental comments to the article.
